# *Mitragyna speciosa* Korth Leaf Pellet Supplementation on Feed Intake, Nutrient Digestibility, Rumen Fermentation, Microbial Protein Synthesis and Protozoal Population in Thai Native Beef Cattle

**DOI:** 10.3390/ani12233238

**Published:** 2022-11-22

**Authors:** Burarat Phesatcha, Kampanat Phesatcha, Metha Wanapat

**Affiliations:** 1Department of Agricultural Technology and Environment, Faculty of Sciences and Liberal Arts, Rajamangala University of Technology Isan, Nakhon Ratchasima 30000, Thailand; 2Department of Animal Science, Faculty of Agriculture and Technology, Nakhon Phanom University, Nakhon Phanom 48000, Thailand; 3Tropical Feed Resources Research and Development Center (TROFREC), Department of Animal Science, Faculty of Agriculture, Khon Kaen University, Khon Kaen 40002, Thailand

**Keywords:** *Mitragyna speciosa* Korth, supplementation, rumen enhancer, ruminants

## Abstract

**Simple Summary:**

The leaves of *Mitragyna speciosa* Korth contain phytogenic compounds, including total phenolics, flavonoids and mitragynine. The objective of this study was to evaluate the effects of *Mitragyna speciosa* Korth leaf pellets (MSLP) on feed intake and nutrient digestibility in Thai native beef cattle. The findings suggest that supplementation of MSLP at 10–30 g/hd/d significantly increased nutrient digestibility, VFA concentration, and especially propionic acid and microbial protein synthesis while decreasing protozoal populations and estimated CH_4_ production.

**Abstract:**

This experiment evaluated the use of *Mitragyna speciosa* Korth leaf pellets (MSLP) on feed intake and nutrient digestibility in Thai native beef cattle. Four Thai native beef cattle steers were randomly assigned according to a 4 × 4 Latin square design to receive four dietary treatments. The treatments were as follows: control (no supplementation), MSLP supplement at 10 g/hd/d, MSLP supplement at 20 g/hd/d and MSLP supplement at 30 g/hd/d, respectively. All animals were fed a concentrate mixture at 0.5% body weight, while urea lime-treated rice straws were fed ad libitum. Findings revealed that feed intakes were increased by MSLP, which also significantly increased the digestibility of dry matter (DM), organic matter (OM) and neutral detergent fiber (NDF). Ruminal total volatile fatty acid (TVFA) concentration and propionate (C_3_) proportion were increased (*p* < 0.05) with MSLP supplementation, whereas ruminal ammonia-N (NH_3_-N), plasma urea nitrogen (PUN), acetate (C_2_), C_2_:C_3_ ratio and estimated methane (CH_4_) production decreased (*p* < 0.05). Total bacterial, *Fibrobacter succinogenes* and *Ruminococus flavefaciens* populations increased (*p* < 0.05) at high levels of MSLP supplementation, while protozoal populations and methanogenic archaea reduced (*p* < 0.05). Supplementation of MSLP also increased the efficiency of microbial nitrogen protein synthesis. Supplementing beef cattle with MSLP 10–30 g/hd/d significantly increased rumen fermentation end products and nutrient digestibility by mitigating protozoal populations and estimated CH_4_ production.

## 1. Introduction

Farm animal welfare has become of great interest to consumers within many regions. Appropriate stock–people training should be conducted to improve the human-animal relationship with positive effects on animal welfare, productivity and stock–people safety [1]. Agricultural livestock production ranks among the most environmentally impactful industry sectors at the global level, and within the livestock sector, beef production accounts for a large proportion of environmental damage [2]. Ruminant production is associated with poor animal welfare, biodiversity loss, land-use change, human illness due to zoonosis, ill health due to increased consumption of livestock-derived foods and increased resistance of pathogenic microbes as a result of antibiotic feed additives [3,4]. Rumen microbial fermentation produces methane (CH_4_), a significant greenhouse gas that contributes to global warming. Ruminal CH_4_ production, in addition to its adverse effect on the environment, represents a loss of 3% to 10% of the gross energy intake of the animal. Herbal plants containing phytogenic compounds are used to enhance health and treat ailments [5], including anthelmintic and acaricidal disorders, as well as surgical and gynecological treatments and bovine mastitis [6,7]. Phytogenic compounds such as condensed tannins (CT) and saponins (SP) are nutritional components found in plants that benefit animal health by increasing cattle productivity and reducing CH_4_ emissions from ruminants [8]. Plant secondary bioactive metabolites can reduce methane synthesis in the rumen either directly or indirectly by decreasing methanogens or protozoal populations [9]. Matra et al. [10] found that the addition of *Hylocereus undatus* peel powder containing CT and SP to feed resulted in a lower protozoal population and CH_4_ production. Currently, local feed additives are attracting increased attention, particularly those made from fruit peel that can be found on farms or in commercial canning facilities. Fruit peel in powder or pellet form can be used to supplement cattle and buffaloes’ feed. Fruit peel of rambutan (*Nephelium lappaceum* L.) and mangosteen (*Garcinia mangostana*) produced noteworthy results [11] caused by higher levels of rumen fermentation while byproducts such as C_3_, buffered rumen pH and improved microbial protein synthesis, thereby reducing methane production [11,12]. Phesatcha et al. [13] found that CT and SP in *Flemingia macrophylla* pellets had a significant impact on the efficiency of nutrient consumption, rumen fermentation byproducts and milk production in lactating dairy cows. Lower methanogenic populations in Paulownia leaves with high bioactive metabolite content reduced methane generation without changing substrate degradability or volatile fatty acid concentrations [14]. Wallace et al. [15] suggested that saponins disrupted protozoa by forming compounds with sterols on the protozoal membrane surface, causing protozoa impairment and disintegration.

*Mitragyna speciosa* Korth is a natural herbal source in Southeast Asia that has been used to treat opioid abuse and withdrawal symptoms in humans [16]. *Mitragyna speciosa* Korth trees can grow 4–9 m high with a width of 5 m. The tree thrives in healthy, moist soil with moderate to full sun exposure. The leaves and smaller stems are consumed by local people as a vegetable [17], with historical use by Thai and Malay people to treat opiate addiction as well as pain and weariness [18]. Recently, *M. speciosa* was delisted in Thailand as an illegal substance. The leaves of *M. speciosa* have stimulant properties similar to cocaine in low doses and sedative properties similar to opioids in high doses. Preclinical research has identified antioxidant, antibacterial, antiproliferative, anti-inflammatory and antinociceptive properties of *M. Speciosa* [9]. The leaves of *M. speciosa* contain bioactive secondary metabolites from a variety of phytochemical categories, including total phenolics 407.83 ± 2.50 GAE mg/g (Gallic acid equivalent, GAE) and flavonoids 194.00 ± 5.00 QE mg/g (Quercetin equivalent, QE), as well as mitragynine content ranging from 6.53 to 7.19% [16,19]. It was hypothesized that supplementation of *M. speciosa* leaf pellets (MSLP) could improve nutrient digestibility and rumen fermentation efficiency while reducing protozoa populations and CH_4_ production. The impacts of phytogenic compounds such as CT and SP in MSLP on microbial populations and CH_4_ production in Thai native beef cattle have not been previously studied. Therefore, the objective of the study was to evaluate the effects of adding CT and SP from MSLP on rumen fermentation and nutrient degradability while also estimating the reduction of CH_4_ production and microbial protein synthesis in Thai native beef cattle.

## 2. Materials and Methods

### 2.1. Ethical Procedure

The study design and plan strictly followed the norms of the Animal Ethics Committee of Rajamangala University of Technology Isan, Nakhon Ratchasima, Thailand (4/2565; 28 April 2022). According to Thailand’s National Research Council’s Ethics of Animal Experimentation, approval was required to collect rumen fluid from animals for this study’s main objective, which comprised laboratory examination of ruminant feeds.

### 2.2. Dietary Treatments and Experimental Design

Four Thai native beef cattle with a 128 ± 10 kg live weight were randomly assigned according to a 4 × 4 Latin square design to receive four dietary treatments. The treatments were as follows: control (no supplementation), MSLP supplement at 10 g/hd/d, MSLP supplement at 20 g/hd/d and MSLP supplement at 30 g/hd/d, respectively. All animals were fed a concentrate mixture at 0.5% body weight, while urea lime-treated rice straws were fed ad libitum. Clean, fresh water and mineral blocks were available at all times. Ingredient compositions of concentrate mixtures and nutrient compositions are presented in Table 1. The animals had 10 mL injections of vitamins A, D_3_ and E before the treatments were administered, and they were also doused with anthelmintics before the experiment. The rice straw (*Oryza sativa* L.) (RS) was treated with 2.0% urea + 2.0% lime (ULTRS) by adding 2 g urea and 2 g lime in 100 mL of water to 100 g (91% DM) and covered with a sheet of plastic for at least 10 days [20].

The study was carried out across four periods, with each period lasting for 21 days. The first 14 days were used for the adaptation period and for feed dry matter intake measurements, while the last 7 days were for sample collection (feeds, feces and urine).

### 2.3. Sample Collection and Chemical Analyses

Prior to chemical analysis, samples of feeds and refusals were composited by period from daily samples taken during the collection period. Collection of feces and urine from each animal was performed in the morning and in the afternoon. About 200 g of total fresh weight of feces samples were obtained through rectal sampling, while about 50 mL of urine samples were obtained by spot sampling. By manually stimulating the penis, the urine of each animal was collected. Following oven drying and grinding (1 mm screen using a Cyclotech Mill, Tecator, 1093, Hoganes, Sweden), composite samples were analyzed for DM, ash, and CP content [21] as well as acid insoluble ash (AIA). To calculate the digestibility of nutrients, AIA was utilized as an internal indigestible flow marker [22]. According to Van Soest et al. [23], they were examined for acid detergent fiber (ADF) and neutral detergent fiber (NDF).

Blood and rumen fluid samples were obtained on the final day of each study period at 0 and 4 h following the morning feed; each time, a stomach tube attached to a vacuum pump removed about 200 mL of rumen fluid from the rumen. The rumen fluid was immediately measured for pH and temperature using portable pH meter (HANNA Instrument HI 8424 microcomputer, Hanna Instruments (S) Pte Ltd., 161 Kallang Way, Singapore). Rumen fluid samples were then filtered through four layers of cheesecloth and divided into 2 portions. The first portion was used for the analysis of NH_3_-N and volatile fatty acids (VFA), where 5 mL of 1M H_2_SO_4_ was added to 45 mL of rumen fluid and then centrifuged at 1600× *g* for 15 min. About 20 mL of supernatant were collected and analyzed for NH_3_-N analysis [21] and volatile fatty acids (VFA) analysis by high-performance liquid chromatography (HPLC; Model Water 600; UV detector, Millipore Corp., Milford, MA, USA.) [24]. Estimation of ruminal CH_4_ production was performed based on VFA proportions [CH_4_ production = 0.45 (acetate, mol/100 mol) − 0.275 (propionate, mol/100 mol) + 0.4 (butyrate, mol/100 mol) according to Moss et al. [25]. The Koike and Kobayashi [26] approach was used to extract community DNA from rumen fluid. Columns from the QIAgen DNA Mini Stool Kit were used to purify the DNA (QIAGEN, Valencia, CA, USA). Real-time quantitative PCR tests were conducted using template DNA, power SYBR Green PCR Master Mix (Applied Biosystems, Warrington, UK), forward and reverse primers and genomic DNA that had been isolated. According to Edwards et al. [27], specific primers were employed to quantify the microbial population of total bacteria, *Fibrobacter succinogenes*, *Ruminococcus flavefaciens*, *Ruminococus albus* [28], protozoa [28] and methanogenic archaea [29]. The DNA guidelines A Chromo 4TM system was used to determine real-time PCR amplification and detection (Bio-Rad, Hercules, CA, USA).

In addition to collecting rumen fluid, a blood sample (about 10 mL) was also taken from the jugular vein and placed into tubes containing 12 mg of EDTA. The plasma was then separated by centrifuging at 500× *g* for 10 min (Table Top Centrifuge PLC-02, RE Science Inc., Tustin, CA, USA) and kept at −20 °C until plasma urea nitrogen (PUN) analysis [30]. Allantoin and creatinine concentrations in urine were analyzed by HPLC, based on the Chen and Gomes relationship [24], and the quantity of microbial purines ingested was determined from purine derivative excretion. Microbial crude protein (MCP) (g/d) = 3.99 × 0.856 × mmol of purine derivatives excreted was determined by the method of Galo et al. [31]. Efficiency of microbial N supply (EMNS), g/kg of OM digested in the rumen (OMDR) = [(MCP (g/d) × 1000)/DOMR (g)], assuming that rumen digestion = 65% of digestion in total tract.

### 2.4. Statistical Analysis

All data from the experiment were analyzed according to a 4 × 4 Latin square design of treatments and the Statistical Analysis System Institute’s General Linear Models procedures [32]. The statistical model included terms for the animal, period and MSLP supplementation. Data were analyzed using the model:Yijk = l + Mi + Aj + Pk + Bt + eijkt
where Yijk is the observation from animal j, receiving diet i, in period k; l is the overall mean; Mi is the effect of treatment (i = 1–4); Aj is the effect of animal (j = 1–4); Pk is the effect of period (k = 1–4); Bt is the effect of time (t = 1–2) and eijkt is the residual effect. Significance was declared at *p* < 0.05. Results are presented as mean values with the standard error of the means. Differences between treatment means were determined by Duncan’s New Multiple Range test, and differences among means with *p* < 0.05 were represented as statistical differences. Orthogonal polynomials for diet responses were determined by linear and quadratics effects.

## 3. Results and Discussion

### 3.1. Chemical Composition of Experimental Feeds

The nutritional value of the concentrate was 92.7, 91.3, 14.3, 29.1 and 14.6% for DM, OM, CP, NDF and ADF, respectively (Table 1). The nutritive values of the MSLP were 89.4, 85.4, 94.3, 19.8, 50.1 and 26.5% for DM, OM, CP, NDFand ADF, respectively. The phytogenic compounds concentrations in MSLP, particularly mitragynine, CT and SP, were 14.6, 12.1 and 8.2%, respectively, which were equivalent to those reported by Goh et al. [14]; *M. speciosa* Korth leaf included a total phenolic of 407.83 ± 2.50 GAE mg/g, flavonoids 194.00 ±5.00 QE mg/g and mitragynine content of 6.53–7.19%. Furthermore, Chanchula et al. [18] found that *M. speciosa* leaf contained CT and SP are 8.28 and 5.21%, respectively. However, the nutritional profile of *M. speciosa* leaf could shift depending on a number of circumstances, including genetics and the surrounding environment [19].

### 3.2. Dry Matter Intake and Nutrients Digestibility

Supplementation of MSLP increased DM intake while digestibility of DM, OM, CP, NDF and ADF also increased (*p* < 0.05) (Table 2). These findings agreed with Chanjula et al. [18], who found improved total DM intake and apparent digestibility in goats due to the stimulatory effect on microbial growth that improved digestibility afterward. The addition of MSLP improved nutrient digestibility, attributed to rumen microbes being stimulated to a greater rate of feed digestion. The addition of MSLP also increased OM digestibility, and the advantages increased as dietary NDF and ADF concentration increased but did not change the digestibility of CP.

Phytogenic compounds have an impact on the rumen microbiota, which is in control of digesting fiber, producing methane, and biohydrogenating unsaturated fatty acids [3]. In comparison to low or medium doses, high concentrations of CT and SP considerably lower voluntary feed intake and animal performance. This is based on the variability of the dietary phytogenic compounds content [33,34]. Yang et al. [4] found that when CT was added to the diet of beef cattle, it reduced the amount of CH_4_ that was released from the rumen, the relative abundance of protozoa, methanogens and *Ruminococcus albus* to the overall bacterial 16S rDNA, as well as the digestibility of DM, OM and CP. The CT complexes were not fully disassociated from the abomasum, which decreased the digestibility of the nutrients in the lower digestive tract. Additionally, tannins have the ability to bind to dietary proteins, minerals, and polysaccharides like starch, cellulose, hemicellulose and pectin [35,36]. Matra et al. [37] reported that the rumen fermentation end products and nutrient degradability improved with the addition of *Hylocereus undatus* peel powder that included both CT and SP. Similar results were also reported by Wanapat et al. [11] and Phesatcha et al. [38].

### 3.3. Ruminal Parameters and Blood Metabolite

All treatments had pH levels ranging from 6.6 to 6.8. Ammonia nitrogen (NH_3_-N) is the primary nitrogen source for microbial protein synthesis. The NH_3_-N content ranged from 11.9 to 15.6 mg/dL (Table 3). In the current study, NH_3_-N concentration decreased (below 15 mg/dL) as MSLP supplementation increased. At the same time, intake, digestibility and microbial crude protein synthesis increased with increasing MSLP supplementation [11,39,40].

This finding was consistent with that of Phesatcha et al. [19] and Yang et al. [4], who demonstrated that CT had positive effects on nutrition by producing a protein–tannin complex, reducing the availability of ruminal feed protein breakdown and reducing NH_3_-N production. On the other hand, Wanapat et al. [11], Matra et al. [8] and Ampapon et al. [40] found that increasing the supplementation of fruit peel pellets containing CT and SP enhanced NH3-N concentration values. Strategic CT- and SP-containing feed addition could improve rumen efficiency by maintaining higher pH, increasing NH_3_-N concentration, and enhancing microbial protein synthesis. The concentrations of PUN were low and were affected by MSLP supplementation. The level of NH_3_ production in the rumen is closely related to the level of PUN [13,18]. The formation of protein-tannin complexes is also likely to be responsible for the decrease in PUN concentration for the Thai native beef cattle receiving MSLP in this study, as lower PUN concentrations have previously been associated with the inclusion of CT and SP in the diet [41].

The total VFA concentrations in all treatments varied from 91.1 to 102.5 mM. At higher MSLP supplementation levels, the propionate (C_3_) proportion was higher, and acetate concentrations were lower (*p* < 0.05) (Table 3). As a result, rumen methane production was greatly lowered by calculation, and protozoal numbers were reduced. Moreover, feed containing CT and SP had a more dramatic effect in enhancing rumen fermentation by increasing rumen propionic production, decreasing protozoa and, thereby, reducing methane production [4,40,42,43]. Phesatcha et al. [13] revealed that with the supplementation of *Flemingia macrophylla*, the proportion of C_3_ was enhanced, whereas the proportion of C_2_ was decreased. The expected change from C_2_ to C_3_ in the VFA profile was related to the shift from CH_4_ to H_2_, which is beneficial for the energy supply to the host.

### 3.4. Microbial Population and Microbial Protein Synthesis

Supplementation of MSLP significantly increased (*p* < 0.05) total bacteria, *Fibrobacter succinogenes* and *Ruminococus albus*. Additionally, the presence of phytogenic compounds like CT and SP in the MSLP may have contributed to the reduction of protozoa, methanogen archaea, and CH_4_ production in the rumen following MSLP supplementation (*p* < 0.05) (Table 4).

Table 5 displays data on urinary purine derivatives and other relevant indicators. Values of urinary purine derivatives significantly improved (*p* < 0.05) with MSLP supplementation. Microbial and nitrogen synthesis (MNS) and efficiency of microbial nitrogen synthesis (EMNS) were also significantly enhanced (*p* < 0.05) by the level of MSLP supplementation. The highest microbial protein synthesis in terms of quantity and effectiveness was achieved with a high level of MSLP supplementation. Supplying mangosteen peel containing CT and SP to dairy cows increased the efficiency of microbial protein production [44], while decreased protozoal populations enhanced the absorption of dietary nitrogen and boosted the flow of microbial protein production in the intestine [8,28]. Condensed tannins (17.2%) and saponins (10.9%) from mangosteen peel fed to swamp buffaloes at 100 g/head/day enhanced the population of all bacteria and *Ruminococus flavefaciens* while decreasing methanogens (*p* < 0.05) [45]. Tannins decrease methane synthesis in the rumen either directly or indirectly by suppressing methanogens or protozoal populations, respectively. Ku-Vera et al. [36] reported that condensed tannins have a direct impact on rumen methanogenic archaea by binding proteinaceous adhesin or components of the cell envelope, thereby preventing the formation of methanogen-protozoa complexes, lowering interspecies hydrogen transfer and inhibiting methanogen growth [5,46]. Most saponins have an effect on protozoa by forming complexes with the lipid membrane of bacteria, thereby increasing their permeability and ultimately leading to the lysis of microorganisms [47].

Furthermore, the presence of phytogenic compounds in feeds, whether in natural form or as plant extracts, has been shown to have an impact on rumen reduction of methane production, engaged by rumen microbes, as well as having a direct effect on selective microorganisms like protozoa and cellulolytic bacteria, especially in increasing *Fibrobacter succinogenes* and *Butyrivibrio fibrisolvens* and decreasing *Ruminococus flavefaciens* [19,48,49]. According to Phesatcha et al. [12], adding *Garcinia mangostana* peel, which includes CT and SP, revealed decreased protozoa and methanogen populations, as well as reduced CH_4_ production in swamp buffalo.

## 4. Conclusions

Supplementation of MSLP at 10–30 g/hd/d significantly increased nutrient digestibility, propionic acid proportion and microbial protein synthesis while decreasing protozoal populations and estimated CH_4_ production in Thai native beef cattle fed on a urea-calcium hydroxide-treated rice straw-based diet.

## Figures and Tables

**Table 1 animals-12-03238-t001:** Compositions of concentrate mixtures, urea- and calcium hydroxide-treated rice straw, and *Mitragyna speciosa* Korth leaf pellet.

Items	Concentrate	Urea- and Calcium Hydroxide-Treated Rice Straw	*Mitragyna speciosa* Korth Leaf Pellet

Concentrate ingredients, % as fresh basis			
Cassava chip	66.0		
Leucaena leaf meal	18.0		
Rice bran	10.0		
Urea	2.5		
Molasses	2.0		
Mineral mixture	0.5		
Salt	0.5		
Sulfur	0.5		
*Mitragyna speciosa* Korth leaf	-		90.0
Cassava chip starch	-		10.0
Total	100.0		
Chemical composition			
Dry matter, %	92.7	55.1	89.4
	--------------------% of dry matter--------------------		
Organic matter	91.3	88.7	85.4
Ash	8.7	11.3	14.6
Crude protein	14.3	5.6	19.8
Neutral detergent fiber	29.1	72.8	50.1
Acid detergent fiber	14.6	45.9	26.5
Mitragynine	-	-	7.9
Condensed tannins	-	-	13.4
Saponin	-	-	11.2

**Table 2 animals-12-03238-t002:** Effect of *Mitragyna speciosa* Korth leaf pellet supplementation on voluntary feed intake and nutrient digestibility in Thai native beef cattle.

Items	MSLP ^1^ (g/hd/d)				SEM	Contrast	
	0	10	20	30		Linear	Quadratic
Dry matter intake							
kg/day	2.6 ^a^	2.7 ^a^	3.2 ^b^	3.4 ^b^	0.18	0.039	0.041
g/kg BW^0.75^	85.4	86.8	88.3	89.1	1.21	0.069	0.571
Estimate energy intake							
ME (MJ/d)	46.4	47.1	49.3	50.4	0.33	0.053	0.071
ME (MJ/kgDM)	10.4	10.5	10.7	10.7	0.47	0.080	0.092
Nutrient digestibility, %							
Dry matter	63.4 ^a^	64.6 ^a^	67.5 ^b^	69.1 ^b^	0.03	0.048	0.051
Organic matter	63.1 ^a^	65.2 ^b^	66.8 ^b^	68.3 ^c^	0.07	0.043	0.062
Crude protein	55.1	55.7	56.1	57.5	0.14	0.059	0.78
Neutral detergent fiber	61.8 ^a^	62.7 ^ab^	63.9 ^b^	64.2 ^b^	0.11	0.041	0.044
Acid detergent fiber	47.1 ^a^	50.2 ^ab^	52.1 ^b^	53.0 ^b^	0.08	0.042	0.045

**^1^** MSLP = *Mitragyna speciosa* Korth Leaf pellet; SEM = standard error of the mean. Different superscript letters within a column indicate statistical differences.

**Table 3 animals-12-03238-t003:** Effect of *Mitragyna speciosa* Korth leaf pellet supplementation on fermentation characteristics and plasma urea nitrogen in Thai native beef cattle.

Items	MSLP ^1^ (g/hd/d)				SEM	Contrast	
	0	10	20	30		Linear	Quadratic
Ruminal pH	6.6	6.8	6.7	6.7	0.26	0.068	0.093
Temperature, °C	39.1	39.3	39.2	39.1	0.19	0.283	0.425
NH_3_-N, mg/dL	15.6 ^b^	14.3 ^b^	12.1 ^a^	11.9 ^a^	0.31	0.041	0.045
PUN, mg/dL	13.4 ^b^	12.7 ^b^	11.6 ^a^	11.8 ^a^	0.42	0.064	0.083
Total VFAs, mmol/L	91.1 ^a^	95.3 ^b^	99.7 ^c^	102.5 ^c^	2.53	0.041	0.046
VFAs, mol/100 mol							
Acetic acid (C_2_)	68.3 ^b^	67.5 ^b^	65.6 ^a^	64.8 ^a^	0.59	0.033	0.042
Propionic acid (C_3_)	20.2 ^a^	22.8 ^b^	24.9 ^c^	26.7 ^d^	0.46	0.025	0.032
Butyric acid (C_4_)	11.5	9.7	9.5	8.5	0.32	0.058	0.069
C_2_:C_3_	3.38	2.96	2.63	2.43	0.11	0.183	0.314
CH_4_ (mM)	29.6 ^c^	27.8 ^b^	26.4 ^ab^	25.1 ^a^	0.73	0.022	0.035

**^1^** MSLP = *Mitragyna speciosa* Korth Leaf pellet; SEM = standard error of the mean; NH_3_-N = ammonia nitrogen; PUN = plasma urea nitrogen; VFAs = volatile fatty acids; CH_4_ = methane production calculated according to Moss et al. [22]. CH_4_ = 0.45 (C_2_) − 0.275 (C_3_) + 0.4 (C_4_). Different superscript letters within a column indicate statistical differences.

**Table 4 animals-12-03238-t004:** Effect of *Mitragyna speciosa* Korth leaf pellet supplementation on microbial population in Thai native beef cattle.

Items	MSLP ^1^ (g/hd/d)				SEM	Contrast	
	0	10	20	30		Linear	Quadratic
Copies/mL of rumen content,							
Total bacteria, ×10^10^	2.1 ^a^	3.9 ^b^	5.3 ^c^	6.1 ^c^	1.67	0.035	0.069
*F. succinogenes*, ×10^8^	2.7 ^a^	3.1 ^a^	4.6 ^b^	5.8 ^c^	0.85	0.033	0.042
*R. flavafaciens*, ×10^8^	1.9 ^a^	2.2 ^a^	3.8 ^b^	5.6 ^c^	1.63	0.040	0.055
*R. albus*, ×10^8^	3.2	3.9	2.7	3.0	0.18	0.039	0.048
Methanogens, ×10^7^	5.2 ^c^	4.0 ^b^	2.1 ^a^	1.5 ^a^	0.31	0.032	0.061
Protozoa, ×10^4^	7.4 ^c^	6.1 ^b^	4.3 ^a^	3.9 ^a^	1.15	0.040	0.052

**^1^** MSLP = *Mitragyna speciosa* Korth Leaf pellet; SEM = standard error of the mean. Different superscript letters within a column indicate statistical differences.

**Table 5 animals-12-03238-t005:** Effect of *Mitragyna speciosa* Korth leaf pellet supplementation on urinary purine derivatives (PD) and microbial protein synthesis in Thai native beef cattle.

Items	MSLP ^1^ (g/hd/d)				SEM	Contrast	
	0	10	20	30		Linear	Quadratic
**Urinary purine derivatives (mmol/d)**							
Allantoin excretion	19.1 ^a^	22.4 ^b^	25.6 ^c^	26.1 ^c^	5.51	0.035	0.044
Allantoin absorption	52.9 ^a^	55.6 ^b^	59.4 ^c^	60.1 ^c^	4.22	0.046	0.637
MNS (gN/d)	33.1 ^a^	35.8 ^b^	37.9 ^c^	39.2 ^c^	3.96	0.031	0.045
EMNS (g/kg OMDR)	14.5 ^a^	16.2 ^b^	18.3 ^c^	19.1 ^c^	2.47	0.038	0.056

**^1^** MSLP = *Mitragyna speciosa* Korth Leaf pellet; SEM = standard error of the mean; MNS = microbial nitrogen synthesis; EMNS = efficiency of microbial nitrogen synthesis; OMDR = digestible organic matter apparently fermented in the rumen. Different superscript letters within a column indicate statistical differences.

## Data Availability

Data are available upon a reasonable request.

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
