# Peer review of "Mitragyna speciosa Korth Leaf Pellet Supplementation on Feed Intake, Nutrient Digestibility, Rumen Fermentation, Microbial Protein Synthesis and Protozoal Population in Thai Native Beef Cattle"

_animals, 2022, doi:10.3390/ani12233238_

Round 1

Reviewer 1 Report

The manuscript fits well within the scope of the journal. The Authors have investigated an interesting topic and the theme has been properly described. The objectives of the study were clearly defined.

The Introduction is written concisely and provides sufficient background, However, I suggest reading and citing the following articles: 

-Human-Animal Interactions in Dairy Buffalo Farms https://doi.org/10.3390/ani9050246

- Environmental and biodiversity effects of different beef production systems https://doi.org/10.1016/j.jenvman.2021.112523 

Effects of chickpea in substitution of soybean meal on milk production, blood profile and reproductive response of primiparous buffaloes in early lactation  https://doi.org/10.3390/ani10030515

The design of the manuscript was allows for making reliable conclusions.

Results of the authors on the specific thematic are well presented and thoroughly discussed and data interpretation is appropriate.

No significant limitations have been detected, whereas the paper presents novel and useful findings. The presented collected data have significant practical implications.

In conclusion, I recommend the acceptance of the review for publication after minor correction which is provided within the text.

Author Response

Response to Reviewer 1

---------------------------------------------------------------------------------------------------------------------

- English redaction must be improved. I recommend authors to revise it by a native speaker before resubmission to the journal.

Response: Thanks, we have revised it by a native speaker. Please see the modification in the manuscript.

- The manuscript fits well within the scope of the journal. The Authors have investigated an interesting topic and the theme has been properly described. The objectives of the study were clearly defined.

 Response: Thank you for appreciating our work and also providing valuable suggestions.

- The Introduction is written concisely and provides sufficient background, However, I suggest reading and citing the following articles: 

       - Human-Animal Interactions in Dairy Buffalo Farms https://doi.org/10.3390/ani9050246

       - Environmental and biodiversity effects of different beef production systems https://doi.org/10.1016/j.jenvman.2021.112523 

        - Effects of chickpea in substitution of soybean meal on milk production, blood profile and reproductive response of primiparous buffaloes in early lactation  https://doi.org/10.3390/ani10030515

 Response: Thank you for your valuable suggestions. We have revised it.

  1. Napolitano, F.; Serrapica, F.; Braghieri, A.; Masucci, F.; Sabia, E.; De Rosa, Giuseppe. Human-animal interactions in dairy buffalo farm. Animals. 2019, 9, 246.
  2. Angerer, V.; Sabia, E.; Konig von Borstel, U.; Gauly, M. Environmental and biodiversity effects of different beef production system. Environmen. Manage. 2021, 289, 112523.
  3. Serrapica, F.; Masucci, F.; Romano, R.; Napolitano, F.; Sabia, E.; Aiello, A.; Di Francia, A. Effects of chickpea in substitution of soybean meal on milk production, blood profile and reproductive response of primiparous buffaloes in early lactation. Animals. 2020, 10, 515.

- The design of the manuscript was allows for making reliable conclusions.

 Response: Thank you.

- No significant limitations have been detected, whereas the paper presents novel and useful findings. The presented collected data have significant practical implications.

 Response: Thank you.

In conclusion, I recommend the acceptance of the review for publication after minor correction which is provided within the text.

Response: Thank you for appreciating our work and also providing valuable suggestions. Below, we have tried our best to revise accordingly to your comments, and we hope that this will be sufficient academically and prompt for publication. Thanks!

Reviewer 2 Report

    The study aimed at evaluating the lovastatin (Lv) production by solid-state fermentation (SSF) from selected crop residues, considering the post-fermented residues as feed supplements for ruminants. This is an interesting work.  The "Abstruct" , "Material and Methods" and "Discussion" part are well presented.

    However, what is the real significance of this work? For Lv production, or for supplementary food in animal husbandry ? Whether the post-fermented residues are applicable to large-scale production?  In addition,  I haven't seen the supplementary materials (a download web link or files) in my review report forum, therefore, I do not make any comments on the authenticity of this section in the results parts.   The other detail comments are as follows.

---There are many unnecessary lines between words throughout the text (eg., line 51 "en-hanced",  line 102 sub-cultured). Please removed them.

--- The writing format of "A. terreus strains" is not uniform.  Please confirm.

Author Response

Response to Reviewer 2

---------------------------------------------------------------------------------------------------------------------

- English redaction must be improved. I recommend authors to revise it by a native speaker before resubmission to the journal.

Response: Thanks, we have revised it by a native speaker. Please see the modification in the manuscript.

- Are the results clearly presented (Can be improved)

Response: Thank you for appreciating our work and also providing valuable suggestions. Below, we have tried our best to revise accordingly to your comments, and we hope that this will be sufficient academically and prompt for publication. Thanks!

Reviewer 3 Report

The current study contains novel and exciting information regarding the supplementation of Mitragyna speciosa Korth Leaf Pellet in beef cattle diets to modify ruminal fermentation, rumen microbial protein synthesis, and total-tract nutrient digestibility. In general, the writing was quite poor at times and interpretation of the results seemed to be incorrect/misleading at times. Also, the authors should take some time to describe the limitations of the experiment (i.e. n=4, can't separate effects of CT or SP from MSLP) and identify future areas for research (i.e. compare MSLP with CT and SP at equal dietary concentrations). Several line-by-line comments have been provided to improve the manuscript:

Lines 2-4: Feed utilization and rumen fermentation efficiency are not the appropriate terms. Feed intake, nutrient digestibility, and rumen fermentation end-products were measured.

Line 14: Here and throughout the manuscript, phytonutrients and bioactive secondary metabolites are used interchangeably to describe mitragynine, condensed tannins, and saponin. It is recommended that the authors choose 1 description/term and use that consistently throughout the manuscript. Also, if the authors choose to use "phytonutrient", it is recommended that the term be changed to "phytogenic compound", as nutritional aspects for some phytogenic compounds are questionable.

Line 18: VFA production was not measured. VFA concentrations were measured.

Line 19: Here and throughout the manuscript, CH4 production was not measured, it was estimated using an equation. Therefore, it should always be expressed as "estimated CH4 production".

Lines 24-25: Remove T1-T4 as treatment abbreviations. These abbreviations are not used in tables or the results and discussion.

Line 27: "feed intakes were significantly impacted by MSLP" does not give the reader any meaningful information. Suggest revising to include the increase/decrease and the magnitude of the change.

Line 29: Revise to "Ruminal total volatile fatty acid concentration and propionate proportion"

Line 30: Revise to "ruminal" ammonia-N

Line 30: Here and throughout the manuscript, blood urea nitrogen is not the appropriate term. Blood was centrifuged and plasma was recovered. Therefore, revise all BUN to plasma urea nitrogen.

Line 34: Here and throughout the manuscript, do not use words such as improved or impacted to describe statistical relationships. Use increase and decrease. Also, nitrogen absorption was not measured.

Line 40: It is recommended that the authors re-write the introduction, as there is a lot of irrelevant information and lack of clear focus. First, form a hypothesis for the experiment. Then, work backwards to include relevant information to support that hypothesis.

Lines 41-49: Some of this is a bit extrapolated and off-topic. All information in the introduction should lead up to the hypothesis of the experiment.

Line 95: Add hypothesis

Table 1: Values for urea and molasses in the concentrate are different font size/style. For the "Total" row, 100 should be added in the leaf pellet column. What are the units for mitragynine, condensed tannins, and saponin? Is it % of DM for all of them?

Line 119: Change 3 weeks to 21 days.

Line 120: "The first 14 days were used to adjust the treatment and assess the amount of feed 120 consumed, while the final 7 days were used to collect samples of the feed, feces, and urine". This sentence needs more clarification. How long were animals adapted to treatment? How long was feed intake measured for? I'm assuming that the first 14 days were used for treatment adaptation and that the last 7 days were used to measure feed intake.

Line 124-125: Delete: "During the final week of each phase, feed, feces, and urine samples were taken". This was stated previously.

Line 125-126: More description is needed for urinary and fecal collection. How many samples? On which days? Time interval between samples?

Line 130: Suggest revising "internal indication" to "internal indigestible flow marker"

Line 131: Delete "fiber content like"

Line 142: NH3-N abbreviation was previously described

Line 161: It was already mentioned that urine samples were collected. Suggest revising to "Allantoin and creatinine concentrations in urine were analyzed by HPLC"

Lines 169-176: The current study evaluates 4 levels of MSLP supplementation (0, 10, 20, 30 g/d). It seems more appropriate to analyze data with linear and quadratic contrasts to determine effects of levels of supplementation. Also, rumen samples were collected at 0-h and 4-h after feeding. How was time accounted for in the statistical models?

Line 178-188: Suggest deleting this paragraph. It is not really results (no statistics) and can be described entirely by Table 1.

Line 191-193: I am struggling to understand the mechanism that is being described. Typically, when DMI increases, there is an increase in the rate of ruminal passage and an increase in microbial protein synthesis. However, this is typically accompanied by a small decrease (5-10%) in digestibility. 

Table 2: Please fix the font size for P-values for CP and ADF digestibility. At first look, I thought the P-values were non-significant. Also, add superscripts for CP digestibility.

Line 216-218: Revise this sentence "Ammonia nitrogen (NH3-N) is  the primary nitrogen source for microbial protein synthesis, and bacteria also use NH3-N as their main nitrogen source". It says the same thing twice. 

Line 219-221: "Concentrations of NH3-N ranging 15 to 30 mg/dL improved voluntary feed intake, microbial protein synthesis, nutrient digestibility and rumen ecology, whereas NH3-N deficit decreased bacterial growth rate." I suggest the authors revise this sentence. In the current study, NH3-N concentration decreased (below 15 mg/dL) as MSLP supplementation increased. At the same time, intake, digestibility, and microbial crude protein synthesis increased with increasing MSLP supplementation. It is recommended that the authors describe the decreasing ruminal NH3-N concentration along with protozoal and bacterial data. Because MSLP supplementation decreased protozoa, it increases bacterial growth because protozoa are predatory to bacteria. The increase in bacterial growth consumes more NH3-N, which is why the concentration is lower in the rumen. The results of the current study on ammonia, protozoa, and bacteria in response to MSLP seem to be more similar to studies with saponins than with condensed tannins.

Line 227-234: Like stated above, there is not really any evidence from the results suggesting that condensed tannins from MSLP decreased digestibility and ruminal ammonia-N concentration.

Line 236-237: Kidney damage being responsible for a 1.8 mg/dL difference in plasma urea-N seems unlikely in the current study.

Line 241-242: How is this relevant?

Line 243-245: Table 3 shows total VFA concentration and molar proportions of individual VFA. When describing effects on individual VFA, describe as molar proportion not concentration.

Line 243-252: Once again, the shift to propionate seems characteristic of saponin supplementation and it is not mentioned in the discussion.

Line 252: Change "ruminants' nutritional requirements" to "energy supply to the host"

Line 261: Consider revising MNS from microbial nitrogen supply to microbial nitrogen synthesis

Table 5: How was allantoin absorption measured?

Line 295: It is recommended that the entire conclusions section is re-written. VFA production was not measured, and this sentence "The Thai native beef cattle were fed on urea-calcium hydroxide treated rice straw as a roughage source" is irrelevant.  

Author Response

Response to Reviewer 3

---------------------------------------------------------------------------------------------------------------------

The current study contains novel and exciting information regarding the supplementation of Mitragyna speciosa Korth Leaf Pellet in beef cattle diets to modify ruminal fermentation, rumen microbial protein synthesis, and total-tract nutrient digestibility. In general, the writing was quite poor at times and interpretation of the results seemed to be incorrect/misleading at times. Also, the authors should take some time to describe the limitations of the experiment (i.e. n=4, can't separate effects of CT or SP from MSLP) and identify future areas for research (i.e. compare MSLP with CT and SP at equal dietary concentrations). Several line-by-line comments have been provided to improve the manuscript:

Response: Thank you for appreciating our work and also providing valuable suggestions. Below, we have tried our best to revise accordingly to your comments, and we hope that this will be sufficient academically and prompt for publication. Thanks!

Lines 2-4: Feed utilization and rumen fermentation efficiency are not the appropriate terms. Feed intake, nutrient digestibility, and rumen fermentation end-products were measured.

Response: Thanks, we have revised it.

Line 14: Here and throughout the manuscript, phytonutrients and bioactive secondary metabolites are used interchangeably to describe mitragynine, condensed tannins, and saponin. It is recommended that the authors choose 1 description/term and use that consistently throughout the manuscript. Also, if the authors choose to use "phytonutrient", it is recommended that the term be changed to "phytogenic compound", as nutritional aspects for some phytogenic compounds are questionable.

Response: Thanks, we have revised it.

Line 18: VFA production was not measured. VFA concentrations were measured.

Response: Thanks, we have revised it.

Line 19: Here and throughout the manuscript, CH4 production was not measured, it was estimated using an equation. Therefore, it should always be expressed as "estimated CH4 production".

Response: Thanks, we have revised it.

Lines 24-25: Remove T1-T4 as treatment abbreviations. These abbreviations are not used in tables or the results and discussion.

Response: Thanks, we have revised it.

Line 27: "feed intakes were significantly impacted by MSLP" does not give the reader any meaningful information. Suggest revising to include the increase/decrease and the magnitude of the change.

Response: Thanks, we have revised it.

Line 29: Revise to "Ruminal total volatile fatty acid concentration and propionate proportion"

Response: Thanks, we have revised it.

Line 30: Revise to "ruminal" ammonia-N           

Response: Thanks, we have revised it.

Line 30: Here and throughout the manuscript, blood urea nitrogen is not the appropriate term. Blood was centrifuged and plasma was recovered. Therefore, revise all BUN to plasma urea nitrogen.

Response: Thanks, we have revised it.

Line 34: Here and throughout the manuscript, do not use words such as improved or impacted to describe statistical relationships. Use increase and decrease. Also, nitrogen absorption was not measured.

Response: Thanks, we have revised it.

Line 40: It is recommended that the authors re-write the introduction, as there is a lot of irrelevant information and lack of clear focus. First, form a hypothesis for the experiment. Then, work backwards to include relevant information to support that hypothesis.

Response: Thanks, we have revised it.

Lines 41-49: Some of this is a bit extrapolated and off-topic. All information in the introduction should lead up to the hypothesis of the experiment.

Response: Thanks, we have revised it.

Line 95: Add hypothesis

Response: Thanks, we have revised it.

Table 1: Values for urea and molasses in the concentrate are different font size/style. For the "Total" row, 100 should be added in the leaf pellet column. What are the units for mitragynine, condensed tannins, and saponin? Is it % of DM for all of them?

Response: Thanks, the units for mitragynine, condensed tannins, and saponin,  it is % of DM for all of them.

Line 119: Change 3 weeks to 21 days.

Response: Thanks, we have revised it.

Line 120: "The first 14 days were used to adjust the treatment and assess the amount of feed 120 consumed, while the final 7 days were used to collect samples of the feed, feces, and urine". This sentence needs more clarification. How long were animals adapted to treatment? How long was feed intake measured for? I'm assuming that the first 14 days were used for treatment adaptation and that the last 7 days were used to measure feed intake.

Response: Thanks, we have revised it. The first 14 days were used for the adaptation period and for feed dry matter intake measurements, while the last 7 days were for sample collection (feeds, faces, and urine).

Line 124-125: Delete: "During the final week of each phase, feed, feces, and urine samples were taken". This was stated previously.

Response: Thanks, we have revised it.

Line 125-126: More description is needed for urinary and fecal collection. How many samples? On which days? Time interval between samples?

Response: Thanks, we have revised it “Collection of feces and urine from each animal was done in the morning and in the afternoon. About 200 g total fresh weight of feces samples were obtained through rectal sampling, while about 50 ml urine samples were obtained by spot sampling. By manually stimulating the pennis, the urine of each animal was collected.”

Line 130: Suggest revising "internal indication" to "internal indigestible flow marker"

Response: Thanks, we have revised it.

Line 131: Delete "fiber content like"

Response: Thanks, we have revised it.

Line 142: NH3-N abbreviation was previously described

Response: Thanks, we have revised it.

Line 161: It was already mentioned that urine samples were collected. Suggest revising to "Allantoin and creatinine concentrations in urine were analyzed by HPLC"

Response: Thanks, we have revised it.

Lines 169-176: The current study evaluates 4 levels of MSLP supplementation (0, 10, 20, 30 g/d). It seems more appropriate to analyze data with linear and quadratic contrasts to determine effects of levels of supplementation. Also, rumen samples were collected at 0-h and 4-h after feeding. How was time accounted for in the statistical models?

Response: Thanks, we have revised it.

Line 178-188: Suggest deleting this paragraph. It is not really results (no statistics) and can be described entirely by Table 1.

Response: Thanks, this paragraph is necessary to describe Table 1.

Line 191-193: I am struggling to understand the mechanism that is being described. Typically, when DMI increases, there is an increase in the rate of ruminal passage and an increase in microbial protein synthesis. However, this is typically accompanied by a small decrease (5-10%) in digestibility. 

Response: Thanks, we have revised it “These findings agreed with Chanjula et al. [18]  who found improved total DM intake and apparent digestibility in goats due to the stimulatory effect on microbial growth that improved digestibility afterwards. Addition of MSLP improved nutrient digestibility, attributed to rumen microbes being stimulated to a greater rate of feed digestion. Addition of MSLP also improved OM digestibility, and advantages increased as dietary NDF concentration increased.”

Table 2: Please fix the font size for P-values for CP and ADF digestibility. At first look, I thought the P-values were non-significant. Also, add superscripts for CP digestibility.

Response: Thanks, we have revised it “Addition of MSLP also increased OM digestibility, and advantages increased as dietary NDF and ADF concentration increased but did not change digestibility of CP.”

Line 216-218: Revise this sentence "Ammonia nitrogen (NH3-N) is  the primary nitrogen source for microbial protein synthesis, and bacteria also use NH3-N as their main nitrogen source". It says the same thing twice. 

Response: Thanks, we have revised it.

Line 219-221: "Concentrations of NH3-N ranging 15 to 30 mg/dL improved voluntary feed intake, microbial protein synthesis, nutrient digestibility and rumen ecology, whereas NH3-N deficit decreased bacterial growth rate." I suggest the authors revise this sentence. In the current study, NH3-N concentration decreased (below 15 mg/dL) as MSLP supplementation increased. At the same time, intake, digestibility, and microbial crude protein synthesis increased with increasing MSLP supplementation. It is recommended that the authors describe the decreasing ruminal NH3-N concentration along with protozoal and bacterial data. Because MSLP supplementation decreased protozoa, it increases bacterial growth because protozoa are predatory to bacteria. The increase in bacterial growth consumes more NH3-N, which is why the concentration is lower in the rumen. The results of the current study on ammonia, protozoa, and bacteria in response to MSLP seem to be more similar to studies with saponins than with condensed tannins.

Response: Thanks, we have revised it.

Line 227-234: Like stated above, there is not really any evidence from the results suggesting that condensed tannins from MSLP decreased digestibility and ruminal ammonia-N concentration.

Response: Thanks, we have revised it.

Line 236-237: Kidney damage being responsible for a 1.8 mg/dL difference in plasma urea-N seems unlikely in the current study.

Response: Thanks, we have revised it.

Line 241-242: How is this relevant?

Response: Thanks, we have deleted it.

Line 243-245: Table 3 shows total VFA concentration and molar proportions of individual VFA. When describing effects on individual VFA, describe as molar proportion not concentration.

Response: Thanks, we have revised it.

Line 243-252: Once again, the shift to propionate seems characteristic of saponin supplementation and it is not mentioned in the discussion.

Response: Thanks, we have revised it.

Line 252: Change "ruminants' nutritional requirements" to "energy supply to the host"

Response: Thanks, we have revised it.

Line 261: Consider revising MNS from microbial nitrogen supply to microbial nitrogen synthesis

Response: Thanks, we have revised it.

Table 5: How was allantoin absorption measured?

Response: Thanks, we have revised it. Based on the Chen and Gomes relationship [24], the quantity of microbial purines ingested was determined from purine derivative excretion. Microbial crude protein (MCP) (g/d) = 3.99 × 0.856 × mmol of purine derivatives excreted was determined by the method of Galo et al. [31].

Line 295: It is recommended that the entire conclusions section is re-written. VFA production was not measured, and this sentence "The Thai native beef cattle were fed on urea-calcium hydroxide treated rice straw as a roughage source" is irrelevant.  

Response: Thanks, we have revised it.

Round 2

Reviewer 3 Report

The authors have made several changes to improve the manuscript. Upon reviewing again, I have a few comments for further improvement.

In the title, it is suggested to change "rumen fermentation efficiency" to "rumen fermentation"

Throughout the manuscript, it is recommended that blood urea nitrogen (BUN) is changed to plasma urea nitrogen (PUN) because the reader can interpret these values differently based on the terminology.

In the simple summary, abstract, and conclusions, the authors state that supplementation of MSLP is recommended at 20-30 g/d to improve rumen fermentation, increase digestibility, increase MNS, decrease protozoa, and decrease estimated methane. It is suggested that the authors revise this conclusion to include data with 10 g/d MSLP supplementation because it also improved all of these parameters but to a lesser extent than the 20-30 g/d levels.

Author Response

Response to Reviewer 3

---------------------------------------------------------------------------------------------------------------------

The authors have made several changes to improve the manuscript. Upon reviewing again, I have a few comments for further improvement.

In the title, it is suggested to change "rumen fermentation efficiency" to "rumen fermentation"

Response: Thank you for appreciating our work and also providing valuable suggestions. Below, we have tried our best to revise accordingly to your comments, and we hope that this will be sufficient academically and prompt for publication. Thanks!

We have revised the title “Mitragyna speciosa Korth Leaf Pellet Supplementation on Feed Intake, Nutrient Digestibility and Rumen Fermentation, Microbial Protein Synthesis and Protozoal Population in Thai Native Beef Cattle.”

Throughout the manuscript, it is recommended that blood urea nitrogen (BUN) is changed to plasma urea nitrogen (PUN) because the reader can interpret these values differently based on the terminology.

Response: Thanks, we have revised it throughout the manuscript.

In the simple summary, abstract, and conclusions, the authors state that supplementation of MSLP is recommended at 20-30 g/d to improve rumen fermentation, increase digestibility, increase MNS, decrease protozoa, and decrease estimated methane. It is suggested that the authors revise this conclusion to include data with 10 g/d MSLP supplementation because it also improved all of these parameters but to a lesser extent than the 20-30 g/d levels.

Response: Thanks for providing valuable suggestions. We have revised simple summary, abstract and conclusions.
